# Nonalcoholic Fatty Liver Disease Is Related to Abnormal Corrected QT Interval and Left Ventricular Hypertrophy in Chinese Male Steelworkers

**DOI:** 10.3390/ijerph192114555

**Published:** 2022-11-06

**Authors:** Wei-Chin Hung, Teng-Hung Yu, Cheng-Ching Wu, Thung-Lip Lee, Wei-Hua Tang, Chia-Chi Chen, I-Cheng Lu, Fu-Mei Chung, Yau-Jiunn Lee, Chia-Chang Hsu

**Affiliations:** 1Division of Cardiology, Department of Internal Medicine, E-Da Hospital, Kaohsiung 82445, Taiwan; 2School of Medicine, College of Medicine, I-Shou University, Kaohsiung 82445, Taiwan; 3Division of Cardiology, Department of Internal Medicine, E-Da Cancer Hospital, Kaohsiung 82445, Taiwan; 4School of Medicine for International Students, College of Medicine, I-Shou University, Kaohsiung 82445, Taiwan; 5Division of Cardiology, Department of Internal Medicine, Taipei Veterans General Hospital, Yuli Branch, Hualien 98142, Taiwan; 6Faculty of Medicine, School of Medicine, National Yang Ming Chiao Tung University, Taipei 112304, Taiwan; 7Department of Pathology, E-Da Hospital, Kaohsiung 82445, Taiwan; 8College of Medicine, I-Shou University, Kaohsiung 82445, Taiwan; 9Department of Occupational Medicine, E-Da Hospital, Kaohsiung 82445, Taiwan; 10Lee’s Endocrinologic Clinic, Pingtung 90000, Taiwan; 11Division of Gastroenterology and Hepatology, Department of Internal Medicine, E-Da Hospital, Kaohsiung 82445, Taiwan; 12Health Examination Center, E-Da Dachang Hospital, Kaohsiung 80794, Taiwan; 13The School of Chinese Medicine for Post Baccalaureate, College of Medicine, I-Shou University, Kaohsiung 82445, Taiwan

**Keywords:** corrected QT prolongation, left ventricular hypertrophy, nonalcoholic fatty liver disease, steelworkers

## Abstract

Objectives: Nonalcoholic fatty liver disease (NAFLD) has been associated with an increased risks of corrected QT (QTc) prolongation and left ventricular hypertrophy (LVH), both of which are associated with the development of cardiovascular disease. Rotating night shift work and a higher risk of incident NAFLD have been reported in male steelworkers. This study aimed to investigate the association of the severity of NAFLD with a prolonged QTc interval and LVH in a large cohort of Chinese male steelworkers. Methods: We examined baseline data of 2998 male steel workers aged 26 to 71 years at two plants. All workers at both plants received regular health assessments, including 12-lead ECG and echocardiography. Abdominal ultrasonography was performed to evaluate the severity of NAFLD. QTc prolongation was defined as follows: normal ≤ 430 ms, borderline 431–450 ms, and abnormal ≥ 451 ms. LVH was defined as a left ventricular mass index (LVMI) >131 g/m^2^. Associations of NAFLD with an abnormal QTc interval and LVH were examined using univariate and multivariate analyses. Results: The QTc interval and the LVMI were significantly correlated with the NAFLD fibrosis score, and the severity of NAFLD was correlated with an abnormal QTc interval and LVH (*p* for trend < 0.05). Multivariate analysis showed that in comparison to the workers without NAFLD, the odds ratios of having an abnormal QTc interval and LVH were 2.54 (95% CI: 1.22–5.39, *p* = 0.013) times and 2.23 (95% CI: 1.02–5.01, *p* = 0.044) times higher in the workers with moderate/severe NAFLD. Conclusions: NAFLD may be closely associated with the risks of an abnormal QTc interval and LVH, suggesting that regular electrocardiogram and echocardiogram monitoring could be used to evaluate the risk of arrhythmia and LVH in male steelworkers with NAFLD.

## 1. Introduction

Nonalcoholic fatty liver disease (NAFLD) is the most common liver disease worldwide, with general population rates ranging from 5–40% in the Asia–Pacific region [1,2] and 20–30% in Western countries [3]. The reported prevalence of NAFLD in Taiwan ranges from 11.4 to 41% [4,5], and it is an emerging public health problem. NAFLD is defined as fat accumulation in the liver identified through imaging or histology after ruling out secondary causes (e.g., viral hepatitis, excess alcohol consumption, certain medications, or other medical disorders). It is a complex metabolic disease linked with dyslipidemia, obesity, and insulin resistance [6]. Patients with NAFLD, and especially the more severe nonalcoholic steatohepatitis, have been reported to have an elevated risk of developing cirrhosis [7], of whom 4–27% may develop hepatocellular carcinoma [8]. NAFLD has also been associated with an elevated risk of QT prolongation, atrial fibrillation, and mortality [9,10].

The QT interval is measured from QRS complex initiation to T-wave completion, and it represents the duration of ventricular electrical depolarization and repolarization. A prolonged QT interval indicates an extension of this vulnerable period, and it is a risk factor for malignant ventricular dysrhythmias [11]. Moreover, an excess QT interval prolongation has been linked to sudden cardiac death and ventricular tachycardia. Furthermore, even within a reference range, QT interval duration has been demonstrated to be a predictor of cardiovascular mortality in the general population [12,13,14]. Left ventricular hypertrophy (LVH) is defined as an increase in the mass of the left ventricle, caused by either an enlarged left ventricular cavity, thicker wall, or both. A thicker left ventricular wall is most commonly attributable to chamber dilatation and pressure overload due to volume overload [15]. LVH is usually attributable to factors, such as severe diabetes, high blood pressure, arrhythmias, heart valve problems, and enlargement of the aorta. LVH can lead to severe problems, such as heart failure, ischemic stroke, and sudden cardiac arrest [16]. The QT interval has been associated with metabolic disorders, such as diabetes and obesity, and cardiac disorders, such as coronary artery disease and hypertension [14,17]. In the general population, the prevalence of LVH has been reported to range from 15–20%, with higher rates associated with older age, obesity, and hypertension [15]. A recent study concluded that NAFLD was associated with the risk of cardiac arrhythmia (including heart block, QT prolongation, premature atrial/ventricular contraction, and atrial fibrillation) [18]. Mantovani et al. reported a correlation between NAFLD and LVH independently of other potential confounders and classical cardiovascular risk factors [19].

A higher prevalence of a prolonged QTc interval has been reported in blue-collar workers [20], potentially due to being obese or overweight, smoking, and hypertension. Moreover, a higher risk of LVH has also been reported in blue-collar men [21]. We hypothesized that NAFLD may be associated with the development of a prolonged QTc interval and LVH in blue-collar workers, and if so, strategies aimed at improving NAFLD may reduce their prevalence of QTc interval prolongation and LVH. Therefore, to test our hypothesis, we investigated the role of NAFLD in a cohort of Chinese male steelworkers by examining associations between the severity of NAFLD, electrocardiogram (ECG) parameters, echocardiographic parameters, and other biomarkers.

## 2. Methods

### 2.1. Ethics Statement

The Human Research Ethics Committee of Kaohsiung E-Da Hospital, Kaohsiung, Taiwan approved this cross-sectional study. Written informed consent was obtained from all the enrolled participants.

### 2.2. Study Participants

We recruited steel workers from two plants who attended the Hospital for annual health examinations between 1 January and 31 December 2016. The workers attended E-Da Hospital as it is their official health screening provider. All workers receiving a first examination who did not meet the exclusion criteria as detailed in Section 2.3 were enrolled and then classified into four groups based on the severity of NAFLD as evaluated by abdominal ultrasonography: (1) no (the reference group), (2) mild, (3) moderate, and (4) severe NAFLD.

### 2.3. Exclusion Criteria

The exclusion criteria were patients: (1) <20 years of age; (2) who drank >20 g/day alcohol; (3) with a history of liver cirrhosis; (4) with a history of seropositivity for viral hepatitis, antihepatitis C antibody, or hepatitis B virus surface antigen; (5) missing abdominal ultrasonography data; (6) with atrial flutter or fibrillation, atrioventricular blocks, pacemaker rhythm, or complete left or right bundle-branch blocks; (7) using any known QT interval-altering medication including class I (e.g., flecainide, mexiletine, quinidine, and procainamide) and class III (e.g., amiodarone, vernakalant, and dronedarone) antiarrhythmic medications and tricyclic antidepressants; (8) with inflammatory diseases; and (9) with glucose-altering endocrine disorders including hyperthyroidism and hypothyroidism. Occupational health physicians performed face-to-face interviews during which the above information was confirmed.

### 2.4. Health Examination Protocol

During the health examination, the participants were asked to complete a self-management questionnaire, which asked about the following lifestyle and demographic factors: age, sex, medications, medical history, sleep quality, cigarette smoking and alcohol consumption, physical exercise, health status, and type of employment. Cigarette smoking was classified as a current smoker and a nonsmoker. Physical exercise was assessed using the question “How often did you exercise during the past month?” The response options were: hardly ever, once, and twice or more. Sleep quality was measured using the question “How often did you have poor sleep during the past month?” The response options were: almost never, sometimes, and often or almost always. Those who drank >20 g/day alcohol were defined as having a significant alcohol intake. A trained nurse measured blood pressure with the participant seated using an automated blood pressure measuring device (HEM-907; Omron, Omron, Japan) after a 5 min rest period. The highest measurement in both arms was for analysis.

### 2.5. Anthropometric Data

The weight and height of the participants were measured electronically to 0.1 kg and 0.1 cm, respectively. Waist circumference was measured to within 0.1 cm at the narrowest point between the lowest rib and the uppermost lateral border of the right iliac crest, and hip circumference was measured at the widest point to within 0.1 cm. Body mass index (BMI) was recorded as kg/m^2^.

### 2.6. Laboratory Measurements

All the workers received blood biochemistry tests after fasting for at least 8 h, including glycated hemoglobin (HbA1c), total cholesterol, glucose, triglyceride, high- and low-density lipoprotein cholesterol (HDL-C/LDL-C), and complete blood count. In addition, serum concentrations of aspartate aminotransferase (AST), uric acid, and alanine aminotransferase (ALT) were also measured. A parallel, multichannel analyzer (Hitachi 7170A, Tokyo, Japan) was used for all measurements [22], and an automated cell counter (XE-2100, Sysmex Corp., Kobe, Japan) was used to measure the peripheral leukocyte count. The CKD-EPI study equation was used to assess renal function (estimated glomerular filtration rate [eGFR]) in our male blue-collar workers as follows [23]: GFR = 141 × min(S_cr_/κ, 1)^α^ × max(S_cr_/κ, 1)^−1.209^ × 0.993^Age^, where S_cr_ is serum creatinine (mg/dL); κ is 0.9; α is −0.411, and min/max indicate the minimum/maximum of S_cr_/κ or 1. In addition, the Jaffe method was used to measure serum creatinine.

### 2.7. Definitions

Hypertension was defined as systolic/diastolic blood pressure (SBP/DBP) ≥140/90 mmHg or both or currently receiving antihypertensive agents. Diabetes mellitus (DM) was diagnosed in those with a glycated hemoglobin (HbA1c) level ≥6.5% (48 mmol/mol) or 2 h postprandial glucose level ≥200 mg/dL (11.1 mmol/L) or fasting glucose ≥126 mg/dL (7.0 mmol/L) according to the 2016 American Diabetes Association (ADA) Guidelines [24] or currently receiving antidiabetic agents. In this study, metabolic syndrome was diagnosed in those who met ≥3 of the following criteria: (1) SBP/DBP ≥130/≥85 mmHg, (2) fasting glucose ≥ 100 mg/dL or a diagnosis of DM, (3) serum triglycerides ≥ 150 mg/dL, (4) serum HDL-C <40 mg/dL for men and <50 mg/dL for women, and (5) waist circumference ≥80 cm for women and ≥90 cm for men. We used Taiwan’s Ministry of Health and Welfare definition of obesity rather than the World Health Organization criteria, as the World Health Organization recommended BMI cutoff value for obesity (≥30 kg/m^2^) has been proposed to be too high for Asian populations [25,26]. Therefore, we used the following BMI cutoff values: severe obesity, >35 kg/m^2^; moderate obesity 30 ≤ BMI < 35 kg/m^2^; mild obesity, 27 ≤ BMI < 30 kg/m^2^; overweight, 24 ≤ BMI < 27 kg/m^2^; normal weight, 18.5 ≤ BMI < 24 kg/m^2^; and underweight, <18.5 kg/m^2^ [27]. The stages of eGFR were classified as 1, 2, 3a, 3b, 4, or 5 according to the definition of the 2012 Kidney Disease Outcomes Quality Initiative [28].

### 2.8. Measurements of ECG and QT and QTc Intervals

We performed 12-lead ECG in the morning of the health check-up. A trained nurse performed all ECG examinations, which were conducted for 10 s with the participant in the supine position. QT intervals were manually analyzed by at least two blinded cardiologists. A prolonged QTc interval was defined according to the following cutoff values: abnormal, >450 ms; borderline, 431–450 ms; and normal, ≤430 ms. Detailed information on all measurements is shown in the Appendix A [14,29,30,31,32].

### 2.9. Echocardiographic Measurements

All the participants underwent standard echocardiography during the health check-up using a standardized protocol following the American Society of Echocardiography recommendations, and each variable was analyzed in at least three cycles [33,34]. All measurements were taken by a single experienced physician. LVH was defined as the left ventricular mass index (LVMI) >31 g/m^2^ in our participants [35]. Detailed information on the echocardiographic measurements is shown in the Appendix A.

### 2.10. Abdominal Ultrasonography and NAFLD Fibrosis Score

All the participants underwent abdominal ultrasonography during the health check-up. Two experienced physicians who were blinded to the participants’ clinical data performed the examinations. NAFLD was diagnosed according to increased brightness of the parenchyma compared to the cortex of the right kidney.

The severity of NAFLD was classified into three categories as follows: Severe NAFLD, which was defined as an increase in hepatic brightness, visualization of only the main portal vein walls, and the absence of all smaller portal vein walls. Moderate NAFLD, which was defined as ultrasonographic findings between mild and severe NAFLD. Mild NAFLD, which was defined as an increase in hepatic brightness with a small decrease in the definition of portal vein walls, as previously reported [36].

We calculated the NAFLD fibrosis score using the following formula: −1.675 + 0.037 × age (years) + 0.094 × BMI (kg/m^2^) + 1.13 × IFG/diabetes (yes = 1, no = 0) + 0.99 × AST/ALT − 0.013 × platelet count (× 109/L) − 0.66 × albumin (g/dL).

### 2.11. Statistical Analysis

We used the JMP version 7.0 (SAS Institute, Cary, NC, USA) for the statistical analysis. Categorical data are reported as number (%). The Kolmogorov–Smirnov test was used to test the normality of data. Continuous, non-normally distributed data are presented as median (interquartile range) and continuous normally distributed data as mean ± SD. One-way analysis of variance was used to compare continuous variables among the four study groups, with the four study groups as the categorical variables and each continuous variable as the outcome variable. Continuous variables were tested for trend using a generalized linear model among the four study groups. The studied continuous variables included age, biochemical data, BMI, SBP, DBP, ECG data, echocardiographic data, and differential and total leukocytes counts. The chi-square test was used to compare categorical data among the four study groups, and the Cochran–Armitage test was used to analyze trends.

Univariate and multivariate logistic regression analyses were used to calculate odds ratio (OR) and 95% confidence interval (CI) to evaluate relationships between NAFLD status and the risk of LVH and prolonged QTc interval. Correlations of QTc interval with LVMI and relevant factors were analyzed using Pearson’s correlation coefficients with two-tailed tests of significance. A *p*-value < 0.05 was considered to be statistically significant.

## 3. Results

A total of 2998 male steel workers were included, with a mean LVMI of 92.1 (SD 18.8) g/m^2^, mean QTc interval of 410.5 (SD 22.0) ms, and mean age of 42.8 (SD 7.4) years. A total of 102 participants (3.4%) and 83 participants (2.8%) had an abnormal QTc interval and LVH, respectively. Table 1 lists the baseline characteristics of all participants according to the severity of NAFLD. Moderate/severe NAFLD was associated with higher rates of DM, hypertension, hyperlipidemia, metabolic syndrome, LVH, borderline QTc interval, abnormal QTc interval, hardly ever exercising in the past month, shift work, SBP, DBP, BMI, mild obesity, moderate obesity, and severe obesity. A higher severity of NAFLD was also correlated with a decrease in the normal QTc interval, exercising twice or more in the past month, underweight, normal weight, and overweight.

Table 2 lists the baseline biochemical characteristics, ECG parameters, and echocardiographic parameters of all participants according to the severity of NAFLD. Moderate/severe NAFLD was correlated with higher HbA1c, fasting glucose, total cholesterol, triglycerides, LDL cholesterol, AST, ALT, calcium, uric acid, heart rate, QTc interval, aortic root diameter, left atrial diameter, LVMI, interventricular septum thickness at end-diastole (IVSd), left ventricular internal dimension at end-diastole (LVIDd), left ventricular posterior wall thickness at end-diastole (LVPWd), end-diastolic volume, stroke volume, peak A-wave velocity, and ratio of the left atrial dimension to the aortic annulus dimension (LA/AO). A higher moderate/severe of NAFLD was also associated with a decrease in HDL-C, potassium, end-diastolic volume index (EDVI), end-systolic volume index (ESVI), and ratio of E to A. Furthermore, moderate/severe NAFLD was also associated with higher white blood cell (WBC), neutrophil, monocyte, and lymphocyte counts (*p* for trend <0.0001, Figure 1).

The results of uni- and multivariate statistical analyses for the association between NAFLD status and abnormal QTc interval are shown in Table 3. The workers with mild NAFLD had an elevated risk of an abnormal QTc interval compared to the workers without NAFLD in model 1 and model 2. However, those with mild NAFLD did not have a higher risk of an abnormal QTc interval compared to those without NAFLD in model 3. The participants with moderate/severe NAFLD had an increased risk of an abnormal QTc interval compared to those without NAFLD in models 1 to 3 (OR: 4.09, 95% CI: 2.30–7.57, *p* < 0.0001; OR: 3.69, 95% CI: 2.06–6.84, *p* < 0.0001; and OR: 2.54, 95% CI: 1.22–5.39, *p* = 0.013, respectively).

The results of uni- and multivariate statistical analyses for the association between NAFLD status and LVH are shown in Table 4. The participants with mild NAFLD did not have a higher risk of LVH compared to those without NAFLD in models 1 to 3. The participants with moderate/severe NAFLD had an elevated risk of LVH compared to those without NAFLD in models 1 to 3 (OR: 3.85, 95% CI: 2.02–7.74, *p* < 0.0001; OR: 3.78, 95% CI: 1.98–7.60, *p* < 0.0001; and OR: 2.23, 95% CI: 1.02–5.01, *p* = 0.044, respectively).

Pearson’s correlation analysis was used to investigate correlations among the QTc interval with the LVMI and other relevant parameters. The results show positive correlations between the QTc interval and the LVMI, age, BMI, DBP, SBP, waist circumference, fasting glucose, LDL-C, total cholesterol, HbA1c, triglycerides, creatinine, platelets, NAFLD score, and total WBC, lymphocyte, monocyte, and neutrophil counts and a negative correlation with HDL-C (Table 5). Furthermore, positive correlations were found between the LVMI and BMI, age, DBP, SBP, waist circumference, fasting glucose, HbA1c, triglycerides, creatinine, NAFLD score, and total WBC, monocyte and neutrophil counts, and negative correlations were found between the LVMI and eGFR, hardly ever exercising, and HDL-C (Table 5).

## 4. Discussion

In this study, we analyzed the association of the severity of NAFLD with an abnormal QTc interval and LVH in a large cohort of Chinese male steelworkers and identified three main findings. First, we found associations between moderate/severe NAFLD and an increased QTc interval and LVMI and also elevated WBC, lymphocyte, monocyte and neutrophil counts. Second, moderate/severe NAFLD was associated with an abnormal QTc interval and LVH after adjusting for conventional risk factors including BMI, triglycerides, HDL-C, HbA1c, sodium, potassium, calcium, WBC count and age in a multiple logistic regression analysis. Third, Pearson’s correlation coefficient analysis showed correlations among the QTc interval with the LVMI and other relevant parameters.

Our findings of a positive association between moderate/severe NAFLD and an abnormal QTc interval after adjusting for potential confounding factors are consistent with previous studies [37,38,39,40]. Targher et al. [39] reported an adjusted OR of NAFLD for a prolonged QTc of 2.26 (95% CI 1.4–3.7) in patients with type 2 diabetes. In a study of apparently healthy Taiwan individuals, Hung et al. [38] reported adjusted ORs of severe NAFLD for a prolonged QTc interval of 1.31 (95% CI 1.16–2.24) in women and 1.87 (95% CI 1.16–2.24) in men. In addition, Mangi et al. [40] reported a univariate OR for a prolonged QT interval of 5.09 (95% CI 2.92–8.86) in patients with NAFLD, and Chung et al. [37] reported an adjusted OR for a prolonged QTc interval of 2.05 (95% CI 1.13–3.71) in patients with NAFLD. In the current study, we found an adjusted OR of moderate/severe NAFLD for an abnormal QTc interval of 2.54 (95% CI 1.22–5.39) in male steelworkers, which is similar with the previous studies on NAFLD and type 2 diabetes [37,39] but higher than the general population [38]. Hence, strategies aimed at monitoring the global risk of cardiovascular disease and arrhythmia in blue-collar workers with NAFLD are important.

The mechanisms for the association between NAFLD and an abnormal QTc interval are incompletely understood, especially in blue-collar workers. Traditional cardiometabolic risk factors including smoking, diabetes, lower level of HDL-C, dyslipidemia, and hypertension, in addition to emerging risk factors including ethnicity, inflammatory profile, and abdominal obesity have been positively associated with QTc prolongation [14,41,42,43,44,45,46]. In the present study, we found a significant association of BMI, SBP, DBP, HbA1c, and HDL-C concentrations and the WBC count with the QTc interval. Moreover, the association between the moderate/severe NAFLD and an abnormal QTc interval remained significant after adjusting for these factors.

Another possible mechanism for the association between the prolonged QTc interval and NAFLD may be due to the autonomic imbalance that occurs in NAFLD [47,48]. The duration of the QT interval and QT dispersion is also known to be influenced by the autonomic nervous system [42]. In addition, in an unpublished study, we found that the absolute powers of high- and low-frequency bands were inversely associated with the QTc interval, as evaluated using heart rate variability analysis. These results may support that autonomic modulation affects the QT interval. We did not assess the autonomic imbalance in this study, and further studies are warranted to investigate whether an autonomic imbalance affects the QTc interval among blue-collar workers with NAFLD.

In this study, we also demonstrated that moderate/severe NAFLD was associated with an increase in heart rate, QTc interval, aortic root diameter, left atrial diameter, LVMI, IVSd, LVIDd, LVPWd, end-diastolic volume, stroke volume, peak A-wave velocity, and LA/AO. Furthermore, moderate/severe NAFLD was associated with a decrease in EDVI, ESVI, and ratio of E to A. Moreover, moderate/severe NAFLD was also associated with LVH in male steelworkers after adjusting for the conventional risk factors. These findings are similar to those reported by Mantovani et al., who found that NAFLD was independently associated with LVH in patients with hypertension and type 2 diabetes [19]. Several mechanisms may explain the association between NAFLD and LVH. NAFLD is considered to be a hepatic manifestation of the metabolic syndrome and obesity [44]. The Strong Heart Study found that metabolic syndrome appears to significantly affect the LV mass [49]. Insulin resistance has been demonstrated to induce matrix deposition and cardiomyocyte hypertrophy through complex and multiple mechanisms, regardless of the effects on the systemic blood pressure [50]. Furthermore, people with obesity have been reported to have higher rates of LVH, higher SBP, and higher BMI and more severe nocturnal hypoxemia that have all been independently associated with a higher LV mass. In addition, the effects of elevated blood pressure may amplify the effects of more severe obesity and sleep apnea [51]. Moreover, the low-grade inflammation in NAFLD increases visceral adipose tissue and in turn proinflammatory factors [44]. In the present study, we found associations among the severity of NAFLD with increases in the WBC, neutrophil, lymphocyte, and monocyte counts. Cross-sectional studies have reported associations between LVH and chronic subclinical inflammation [52,53]. In addition, a population-based cohort study [54] found that LVH is itself a proinflammatory factor, and as such that it may lead to an inflammatory state and other chronic conditions including microalbuminuria, chronic renal failure, insulin resistance, and central obesity. In the present study, we also found significant associations among BMI, SBP, DBP, HbA1c, and HDL-C concentrations and the WBC count with the LVMI. The association between the moderate/severe NAFLD and LVH remained significant after adjusting for these factors.

The physical burden of steelworkers might differ according to their duty and task in the factory; however, most of the participants work in shifts, in high temperatures, and in noisy environments. Their task always needs high concentration. Although most of the job is carried by robots and machines, the worker is still in a very stressful physical and mental conditions. Hence, all these occupational-related factors have an impact on the health of steelworkers. Furthermore, at its core, physical labor is simply the tangible stress placed on our bodies, and too much physical labor is not very healthy either. However, healthy and regular physical exercise is not only good human physical health but also mental regulation. Hence, it was necessary that the steelworkers were asked to complete a self-management questionnaire about their regular physical exercise to realize the physical exercise condition for the workers. In the present study, we found that physical exercise (twice or more) can help workers to prevent the development of NAFLD (Table 1). In addition, a previous study showed that physical activity has been associated with decreased LVH [55]. Furthermore, Lecca et al. found that regular physical activity showed a beneficial effect against QTc prolongation [56]. On the contrary, Zhang et al. demonstrated that QT interval duration was not associated with physical activity [57]. In the present study, we did not analyze whether physical work impacted the LVMI and the QT intervals. Further investigations are also needed to investigate whether physical work is also involved and interacts with NAFLD in LVH and the QTc prolongation process in male steelworkers.

There are several limitations to this study. First, we could not determine the causal relationships of NAFLD with an abnormal QTc interval and LVH due to the cross-sectional design of this study. Future studies involving the use of weight loss/exercise to reduce the severity of NAFLD may help elucidate these relationships. Second, all the included subjects were of Chinese ethnicity, which may limit our findings to other populations. Third, NAFLD was diagnosed using abdominal ultrasound and excluding other known causes of chronic liver disease. Although liver biopsy is the gold standard to confirm the diagnosis of NAFLD, hepatic ultrasonography has been shown to have good sensitivity and specificity to detect moderate to severe fatty liver [58]. Fourth, we did not measure homeostatic model assessment-insulin resistance, fasting insulin, genetic factors, adipocytokines, unreported medication use, sympathetic activity, or autonomic imbalance, and therefore their possible confounding effects cannot be ruled out. Finally, our results cannot explain the underlying mechanisms for the associations among NAFLD with an abnormal QTc interval and LVH, and further studies are needed to elucidate this issue.

## 5. Conclusions

We found associations between moderate/severe NAFLD and a prolonged QTc interval and LVH in male steelworkers. Our findings suggest that regular electrocardiogram and echocardiogram monitoring should be performed for steelworkers with NAFLD to identify the presence of LVH, cardiac arrhythmia, and diastolic and systolic dysfunction, all of which may result in an increased risk of cardiac mortality. The benefits of treating the metabolic syndrome and associated NAFLD may also extend to improving cardiovascular health. Hence, we suggest that an integrated strategy considering all potential metabolic syndrome-associated organ complications should be implemented in steelworkers with NAFLD.

## Figures and Tables

**Figure 1 ijerph-19-14555-f001:**
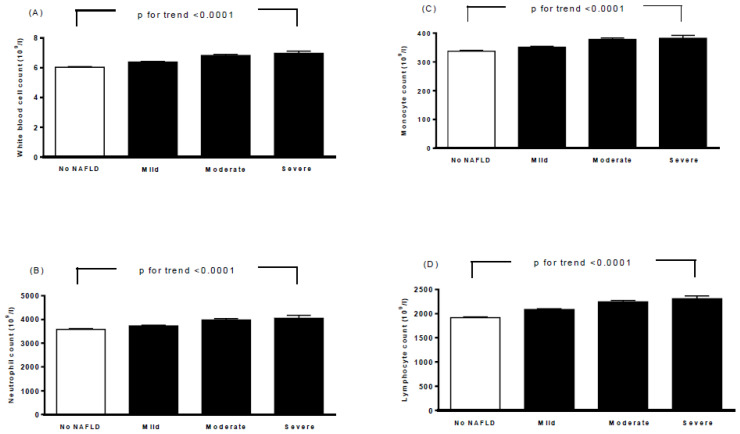
Associations between white blood cell count (**A**), neutrophil count (**B**), monocyte count (**C**), lymphocyte count (**D**), and severity of nonalcoholic fatty liver disease. Bars represent the mean ± SD. A generalized linear model was used to test the trend of each continuous variable across four study groups.

**Table 1 ijerph-19-14555-t001:** Main characteristics according to severity of nonalcoholic fatty liver disease.

		NAFLD				
	No NAFLD	Mild	Moderate	Severe	*p* for Heterogeneity	*p* for Trend
Number	1232	1175	493	98		
Age (years)	41.1 ± 7.6	43.5 ± 7.1	43.5 ± 6.7	42.6 ± 5.7	<0.0001	<0.0001
Diabetes mellitus (n, %)	339 (27.5)	467 (39.7)	229 (46.5)	47 (48.0)	<0.0001	<0.0001
Hypertension (n, %)	294 (23.9)	451 (38.4)	274 (55.6)	63 (64.3)	<0.0001	<0.0001
Hyperlipidemia (n, %)	201 (16.3)	460 (39.2)	290 (58.8)	59 (60.2)	<0.0001	<0.0001
Metabolic syndrome (n, %)	93 (7.6)	309 (26.3)	248 (50.3)	63 (64.3)	<0.0001	<0.0001
LVH (n, %)	25 (2.0)	31 (2.6)	20 (4.1)	7 (7.1)	0.151	0.032
QTc prolongation status (n, %)						
Normal	1100 (89.3)	981 (83.5)	364 (73.8)	62 (63.3)	<0.0001	<0.0001
Borderline	112 (9.1)	152 (12.9)	96 (19.5)	30 (30.6)	<0.0001	<0.0001
Abnormal	21 (1.7)	42 (3.6)	33 (6.7)	6 (6.1)	<0.0001	<0.0001
Current smoker (n, %)	421 (34.2)	404 (34.4)	165 (33.5)	26 (26.5)	0.460	0.341
Physical exercise in the past month (n, %)						
Hardly ever	646 (52.4)	616 (52.4)	295 (59.8)	60 (61.2)	0.013	0.006
Once	365 (29.6)	356 (30.3)	126 (25.6)	28 (28.6)	0.276	0.252
Twice or more	222 (18.0)	203 (17.3)	73 (14.8)	9 (9.2)	0.099	0.025
Poor sleep (n, %)						
Almost never	930 (75.5)	872 (74.2)	379 (76.9)	76 (77.6)	0.666	0.654
Sometimes	201 (16.3)	217 (18.5)	80 (16.2)	14 (14.3)	0.395	0.930
Often or almost always	101 (8.2)	86 (7.3)	35 (7.1)	8 (8.2)	0.799	0.546
Shift work (n, %)	485 (39.4)	525 (44.7)	223 (45.2)	41 (41.8)	0.061	0.045
Systolic BP (mmHg)	119 ± 14	125 ± 16	131 ± 16	134 ± 15	<0.0001	<0.0001
Diastolic BP (mmHg)	75 ± 9	80 ± 10	84 ± 11	87 ± 11	<0.0001	<0.0001
Body mass index (kg/m^2^)	22.7 ± 2.6	25.6 ± 2.8	28.2 ± 3.5	30.4 ± 3.8	<0.0001	<0.0001
Obesity status (n, %)						
Underweight	57 (4.6)	3 (0.3)	1 (0.2)	0 (0.0)	<0.0001	<0.0001
Normal weight	606 (49.2)	193 (16.4)	18 (3.7)	1 (1.0)	<0.0001	<0.0001
Overweight	243 (19.7)	368 (31.3)	86 (17.4)	4 (4.1)	<0.0001	<0.0001
Mild obesity	44 (3.6)	189 (16.1)	117 (23.7)	14 (14.3)	<0.0001	<0.0001
Moderate obesity	7 (0.6)	80 (6.8)	100 (20.3)	35 (35.7)	<0.0001	<0.0001
Severe obesity	274 (22.2)	342 (29.1)	171 (34.7)	44 (44.9)	<0.0001	<0.0001
Chronic kidney disease (n, %)						
G1 (≥90)	202 (16.4)	153 (13.0)	86 (17.4)	15 (15.3)	0.054	0.827
G2 (60–89)	998 (81.0)	992 (84.4)	389 (78.9)	79 (80.6)	0.033	0.708
G3a–G4 (59–15)	31 (2.5)	30 (2.6)	18 (3.7)	4 (4.1)	0.468	0.175

Data are presented as mean ± SD or number (percentage). NAFLD, nonalcoholic fatty liver disease; LVH, left ventricular hypertrophy; LVMI, left ventricular mass index. BP, blood pressure; QTc, corrected QT; Left ventricular hypertrophy defined as LVMI of >131 g/m^2^. Classification of QTc prolongation: normal ≤ 430 ms; borderline 431–450 ms; and abnormal ≥ 451 ms.

**Table 2 ijerph-19-14555-t002:** Biochemical characteristics, ECG parameters, and echocardiographic parameters according to severity of nonalcoholic fatty liver disease.

		NAFLD				
	No NAFLD	Mild	Moderate	Severe	*p* for Heterogeneity	*p* for Trend
Number	1232	1175	493	98		
**Biochemical** **characteristics**						
HbA1c (%)	5.5 ± 0.5	5.7 ± 0.7	5.9 ± 0.9	6.1 ± 1.1	<0.0001	<0.0001
Fasting glucose (mg/dL)	96.9 ± 15.2	101.7 ± 20.0	106.1 ± 28.7	108.3 ± 30.8	<0.0001	<0.0001
Total cholesterol (mg/dL)	186.3 ± 31.9	196.1 ± 32.9	204.1 ± 38.0	204.7 ± 38.6	<0.0001	<0.0001
Triglycerides (mg/dL)	87.0 (64.0–124.0)	128.0 (90.0–185.0)	167.0 (121.0–245.0)	162.5 (121.3–269.3)	<0.0001	<0.0001
HDL cholesterol (mg/dL)	51.0 ± 11.9	44.8 ± 8.8	42.6 ± 7.9	41.9 ± 8.4	<0.0001	<0.0001
LDL cholesterol (mg/dL)	104.7 ± 27.5	115.2 ± 28.9	121.3 ± 31.2	120.8 ± 36.6	<0.0001	<0.0001
AST (U/L)	24.0 (21.0–29.0)	27.0 (23.0–33.0)	34.0 (28.0–42.0)	37.0 (29.0–53.0)	<0.0001	<0.0001
ALT (U/L)	25.0 (19.0–34.0)	35.0 (26.0–48.0)	51.0 (37.0–73.0)	60.5 (43.5–93.3)	<0.0001	<0.0001
Sodium (mEq/L)	140.2 ± 1.6	140.3 ± 1.6	140.3 ± 1.6	140.0 ± 1.8	0.161	0.538
Potassium (mEq/L)	4.02 ± 0.29	4.01 ± 0.27	4.00 ± 0.30	3.95 ± 0.30	0.084	0.024
Calcium (mg/dL)	9.56 ± 0.36	9.55 ± 0.36	9.62 ± 0.34	9.60 ± 0.32	0.002	0.015
Uric acid (mg/dL)	6.1 ± 1.3	6.6 ± 1.3	7.1 ± 1.4	7.5 ± 1.5	<0.0001	<0.0001
eGFR (mL/min/1.73 m^2^)	79.0 ± 11.5	77.7 ± 10.2	78.6 ± 11.1	78.9 ± 12.3	0.028	0.247
Creatinine mg/dL)	1.17 ± 0.63	1.15 ± 0.13	1.15 ± 0.14	1.15 ± 0.15	0.610	0.280
**ECG parameters**						
Heart rate (bpm)	64.9 ± 9.5	66.9 ± 9.3	69.5 ± 9.5	72.2 ± 10.1	<0.0001	<0.0001
PR interval (ms)	157.8 ± 37.4	158.3 ± 20.7	160.4 ± 19.1	158.7 ± 24.8	0.463	0.199
QRS duration (ms)	94.9 ± 13.5	93.7 ± 10.8	94.5 ± 10.6	94.3 ± 10.1	0.137	0.304
QT interval (ms)	393.7 ± 25.5	393.1 ± 24.0	392.0 ± 23.8	388.8 ± 27.6	0.237	0.062
QTc interval (ms)	405.5 ± 23.6	412.0 ± 20.4	419.0 ± 20.3	423.2 ± 20.7	<0.0001	<0.0001
**Echocardiographic** **parameters**						
Aortic root diameter (cm)	3.02 ± 0.39	3.10 ± 0.37	3.18 ± 0.37	3.15 ± 0.35	<0.0001	<0.0001
Left atrial diameter (cm)	3.35 ± 0.42	3.56 ± 0.42	3.72 ± 0.43	3.79 ± 0.36	<0.0001	<0.0001
LVMI (g/m^2^)	89.2 ± 18.8	92.9 ± 18.1	95.4 ± 19.0	96.3 ± 19.8	<0.0001	<0.0001
IVSd (cm)	0.96 ± 0.14	1.00 ± 0.15	1.05 ± 0.15	1.10 ± 0.18	<0.0001	<0.0001
LVIDd (cm)	4.89 ± 0.40	4.95 ± 0.43	5.02 ± 0.41	4.90 ± 0.46	0.002	0.002
LVPWd (cm)	0.87 ± 0.12	0.91 ± 0.12	0.95 ± 0.13	1.01 ± 0.14	<0.0001	<0.0001
End-diastolic volume (mL)	113.5 ± 22.0	116.9 ± 23.2	120.4 ± 22.5	114.9 ± 24.6	0.002	0.001
EDVI (mL/m^2^)	64.7 ± 10.7	62.8 ± 11.1	61.5 ± 10.9	57.1 ± 12.3	<0.0001	<0.0001
End-systolic volume (mL)	36.0 ± 10.1	36.8 ± 11.4	37.7 ± 9.9	36.4 ± 9.7	0.250	0.093
ESVI (mL/m^2^)	20.5 ± 5.3	19.8 ± 5.7	19.3 ± 5.0	18.1 ± 5.1	0.002	0.0002
LVIDs (mm)	3.02 ± 0.34	3.04 ± 0.35	3.07 ± 0.33	3.02 ± 0.34	0.252	0.100
Stroke volume (mL)	77.3 ± 14.9	80.5 ± 15.9	82.7 ± 16.6	78.9 ± 17.6	0.0003	0.0004
SVI (mL/m^2^)	44.1 ± 7.6	43.8 ± 7.8	44.8 ± 24.1	39.9 ± 8.6	0.147	0.514
Fractional shortening (%)	38.5 ± 4.2	38.8 ± 4.5	38.8 ± 4.4	38.5 ± 3.8	0.683	0.476
Ejection fraction (%)	68.4 ± 5.2	68.6 ± 6.0	68.7 ± 5.5	68.4 ± 4.5	0.905	0.560
IVS/LVPW	1.10 ± 0.10	1.10 ± 0.11	1.11 ± 0.09	1.09 ± 0.08	0.577	0.448
Ratio of E to A	1.42 ± 0.41	1.29 ± 0.35	1.20 ± 0.32	1.14 ± 0.35	<0.0001	<0.0001
Peak E-wave velocity (cm/s)	73.5 ± 16.3	71.8 ± 16.1	72.1 ± 16.8	71.9 ± 15.4	0.372	0.180
Peak A-wave velocity (cm/s)	54.0 ± 13.9	58.2 ± 14.5	62.7 ± 15.4	67.3 ± 17.1	<0.0001	<0.0001
LA/AO	1.12 ± 0.17	1.16 ± 0.18	1.18 ± 0.18	1.22 ± 0.16	<0.0001	<0.0001

Data are presented as mean ± SD or median (interquartile range). NAFLD, nonalcoholic fatty liver disease; HDL, high-density lipoprotein cholesterol; LDL, low-density lipoprotein cholesterol; AST, aspartate aminotransferase; ALT, alanine aminotransferase; GFR, glomerular filtration rate; QTc, corrected QT; LVMI, left ventricular mass index; IVSd, interventricular septum thickness at end-diastole; LVIDd, left ventricular internal dimension at end-diastole, LVPWd, left ventricular posterior wall thickness at end-diastole; EDVI, end-diastolic volume index; ESVI, end-systolic volume index; LVIDs, Left ventricular internal dimension at end-systole; SVI, stroke volume index; IVS/LVPW, interventricular septum/Left ventricular posterior wall; and LA/AO, ratio of the left atrial dimension to the aortic annulus dimension.

**Table 3 ijerph-19-14555-t003:** Logistic regression of the association of nonalcoholic fatty liver disease status with abnormal QTc interval.

	Model 1		Model 2		Model 3	
Variables	OR (95%CI)	*p*-Value	OR (95%CI)	*p*-Value	OR (95%CI)	*p*-Value
**NAFLD status**						
No NAFLD	Ref		Ref		Ref	
Mild NAFLD	2.19 (1.24–4.02)	0.006	1.95 (1.10–3.59)	0.022	1.65 (0.89–3.17)	0.112
Moderate/severe NAFLD	4.09 (2.30–7.57)	<0.0001	3.69 (2.06–6.84)	<0.0001	2.54 (1.22–5.39)	0.013

Model 1: Univariate logistic regression analysis. Model 2: Adjusted for age. Model 3: Adjusted for age, body mass index, triglycerides, high-density lipoprotein cholesterol, HbA1c, sodium, potassium, calcium, and white blood cell count. OR, odds ratio; CI, confidence interval; and NAFLD, nonalcoholic fatty liver disease.

**Table 4 ijerph-19-14555-t004:** Logistic regression of the association of nonalcoholic fatty liver disease status with left ventricular hypertrophy.

	Model 1		Model 2		Model 3	
Variables	OR (95%CI)	*p*-Value	OR (95%CI)	*p*-Value	OR (95%CI)	*p*-Value
**NAFLD status**						
No NAFLD	Ref		Ref		Ref	
Mild NAFLD	1.82 (0.96–3.65)	0.068	1.83 (0.96–3.67)	0.066	1.37 (0.69–2.85)	0.372
Moderate/severe NAFLD	3.85 (2.02–7.74)	<0.0001	3.78 (1.98–7.60)	<0.0001	2.23 (1.02–5.01)	0.044

Model 1: Univariate logistic regression analysis. Model 2: Adjusted for age. Model 3: Adjusted for age, body mass index, triglycerides, high-density lipoprotein cholesterol, HbA1c, sodium, potassium, calcium, and white blood cell count. OR, odds ratio; CI, confidence interval; and NAFLD, nonalcoholic fatty liver disease.

**Table 5 ijerph-19-14555-t005:** Pearson’s correlation coefficients between corrected QT interval and left ventricular mass index and relevant parameters in subjects studied.

	Corrected QT Interval			LVMI
Characteristic	r	*p*-Value	r	*p*-Value
Age	0.164	<0.0001	0.195	<0.0001
Body mass index	0.223	<0.0001	0.306	<0.0001
Waist circumference	0.227	<0.0001	0.300	<0.0001
Shift work	−0.001	0.974	0.039	0.133
Hardly ever of physical exercise	0.013	0.515	−0.075	0.004
Systolic blood pressure	0.270	<0.0001	0.292	<0.0001
Diastolic blood pressure	0.275	<0.0001	0.269	<0.0001
Fasting glucose	0.093	<0.0001	0.082	0.001
HbA1c	0.136	<0.0001	0.073	0.004
Total cholesterol	0.079	<0.0001	−0.040	0.122
Triglycerides	0.140	<0.0001	0.062	0.016
HDL cholesterol	−0.088	<0.0001	−0.148	<0.0001
LDL cholesterol	0.050	0.007	−0.016	0.527
Creatinine	0.045	0.015	0.153	<0.0001
Estimated GFR	−0.064	0.090	−0.063	0.014
Albumin	0.024	0.191	−0.019	0.463
Red blood cell count	0.018	0.331	0.048	0.060
Platelets	0.084	<0.0001	−0.034	0.188
Total WBC count	0.126	<0.0001	0.067	0.009
Monocyte count	0.069	0.0002	0.065	0.011
Neutrophil count	0.118	<0.0001	0.060	0.020
Lymphocyte count	0.099	<0.0001	0.036	0.155
NAFLD score	0.039	0.037	0.162	<0.0001
Left ventricular mass index	0.078	0.003	-	-

HDL, high-density lipoprotein; LDL, low-density lipoprotein; GFR, glomerular filtration rate; NAFLD, nonalcoholic fatty liver disease; and LVMI, left ventricular mass index.

## Data Availability

The data presented in this study are available on request from the corresponding author.

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
