# Peer review of "Nonalcoholic Fatty Liver Disease Is Related to Abnormal Corrected QT Interval and Left Ventricular Hypertrophy in Chinese Male Steelworkers"

_ijerph, 2022, doi:10.3390/ijerph192114555_

Round 1

Reviewer 1 Report

 The study is excellent, performed very well, giving interesting, novel results. However, remaining unanswered questions are:

What was the physical burden of steel workers in their everyday work?

Does it make sense to ask the participant in the questionnaire about their regular physical exercise as they were already physical physically active for eight hours? What was the impact of their regular physical work, lasting at least 8 hours a day, upon LVMI and consequently the QT interval?

These considerations could be mentioned in discussion

Author Response

Response to Reviewer 1:

Thank you for reviewer comments and suggestions, the following are our responses:

1. What was the physical burden of steel workers in their everyday work?

Author Response: The physical burden of steelworkers might differ according to their duty and task in the factory, however, most of the participants work in shifts, in high temperatures, and in noisy environments. Their task always needs high concentration. Although most of the job is carried by robots and machines, the worker is still in a very stressful physical and mood condition. Hence, all these occupational-related factors have an impact on the health of steelworkers.

2. Does it make sense to ask the participant in the questionnaire about their regular physical exercise as they were already physically active for eight hours?

Author Response: At its core, physical labor is simply the tangible stress placed on our bodies and too much physical labor isn’t very healthy either. However healthy regular physical exercise not only good human physical health but also mental regulation. Hence, it is necessary that the steelworkers were asked to complete a self-management questionnaire about their regular physical exercise to realize the physical exercise condition for workers. In the present study, we found that physical exercise (twice or more) can help workers to prevent the development of non- alcoholic fatty liver disease (Table 1).

3. What was the impact of their regular physical work, lasting at least 8 hours a day, upon LVMI and consequently the QT interval?

Author Response: A previous study showed that physical activity has been associated with decreased left ventricular hypertrophy (1). Furthermore, Lecca et al. found that regular physical activity showed a beneficial effect against QTc prolongation (2). On the contrary, Zhang et al. demonstrated that QT interval duration was not associated with physical activity (3). In the present study, we did not analyze whether physical work impacted the left ventricular mass index (LVMI) and QT interval. We wish that we could able to figure out the relationship between physical work and LVMI and QT interval and publish it in our future report.

4. These considerations could be mentioned in the discussion.

Author Response: We added sentences and references in the Discussion section to address these points above. Please see page 12, line 382-400 of the revised manuscript.

References:

  1. Kamimura D, Loprinzi PD, Wang W, Suzuki T, Butler KR, Mosley TH, Hall ME. Physical Activity Is Associated With Reduced Left Ventricular Mass in Obese and Hypertensive African Americans. Am J Hypertens. 2017;30:617- 623.
  2. Lecca LI, Portoghese I, Mucci N, Galletta M, Meloni F, Pilia I, Marcias G, Fabbri D, Fostinelli J, Lucchini RG, Cocco P, Campagna M. Association between Work-Related Stress and QT Prolongation in Male Workers. Int J Environ Res Public Health. 2019;16:4781.
  3. Zhang Y, Post WS, Dalal D, Blasco-Colmenares E, Tomaselli GF, Guallar E. Coffee, alcohol, smoking, physical activity and QT interval duration: results from the Third National Health and Nutrition Examination Survey. PLoS One. 2011;6: e17584.

Reviewer 2 Report

A interesting study on a large sample of workers. The self-managed questionnaire maybe a bias for alcohol consumption in that partecipants may tend to lower their consumption? The targeted alcohol consumption is not to high to consider liver disease non-alcoholic? QTc is considered normal when < 450 msec. Instead of considering only the QTc why not considering the 99th percentile? For blood pressure instead of taking in account the highest value, who didm't you considered the average of three consecutives measures?  On echocardiography 5.4 cm for LVEDD, why the choice of this limits? It wasn't possible to work on volume rather than single dimensions?

Author Response

Response to Reviewer 2:

Thank you for the reviewer comments and suggestions, the following are our responses:

  1. The self-managed questionnaire may be a bias for alcohol consumption in that participants may tend to lower their consumption.

Author Response: It is true that for all self-managed questionnaire studies, there are uncertain and biases might occur. However, as it is a self- managed questionnaire, and all the analysis is blind to the researcher (Stated in informed consent to all participants), we believe the participants will more willing to answer the true condition compare to face to face interview visiting study. In the other hand, for the large population study with limited funding, self-managed questionnaire is a more accessible study protocol we could handle.

  1. The targeted alcohol consumption is not too high to consider liver disease non-alcoholic?

Author Response: Nonalcoholic fatty liver disease (NAFLD) is a term for liver conditions affecting people who drink little to no alcohol. In our study, we set participants who drank >20 g/day of alcohol were defined as having significant alcohol intake. It is because if 10g of alcohol (one unit of alcohol drink) is nearly equal to one 350 ml can of regular beer, according to WHO and Taiwan National Health Ministry, consumption of more than 2 units of alcohol could be defined as alcohol overuse.

  1. QTc is considered normal when <450 msec. Instead of considering only the QTc why not consider the 99th percentile?

Author Response: It might be interesting if we use the 99th percentile of QTc in the study. However, because most of the published papers and studies still use the general definition of QTc prolongation (abnormal, >450 ms; borderline, 431-450 ms; normal, ≤430 ms), as a result, we using the traditional QTc prolongation definition in our study (1).

  1. For blood pressure instead of taking into account the highest value, who didn’t you consider the average of three consecutive measures?

Author Response: Thank you for the reviewer’s reminder. Blood pressure are highly affected by emotion, posture, and autonomic activity. It is true that one blood pressure reading is not enough to get an accurate measurement. However, it should be measured over a period of time, not only in one time with several repeated measurements. In some clinical practice suggestions, blood pressure should even be measured at both the beginning and the end of the appointment to gain the correct blood pressure readings. Unfortunately, in large-scale health examinations, it is rather hard to do this. To gain a more accurate blood pressure measurement, in our study, the participant rested for more than 15 minutes while filling out the questionnaire. After then, the participants rested for another 5 minutes and seated comfortably where their arm was relaxed, uncovered, and supported at the level of the heart. Using a appropriate blood pressure cuff fastened at heart level, the blood pressure was measured in both arms. We then choose the highest measure with the aim to minimize the underdiagnosis of hypertension.

  1. On echocardiography 5.4 cm for LVEDD, why is the choice of this limit? It wasn't possible to work on volume rather than single dimensions?

Author Response: Recently, most of heart failure studies used LVEDD but not LV volume as the parameters to observe cardiac size (2). It is because the LV volume is relatively more sensitive to patients’ fluid status, body size, and other factors. As a result, following most of the recent studies (2), we used LVEDD but not LV volume in our study.

References:

  1. Straus SM, Kors JA, De Bruin ML, van der Hooft CS, Hofman A, Heeringa J, Deckers JW, Kingma JH, Sturkenboom MC, Stricker BH, Witteman JC. Prolonged QTc interval and risk of sudden cardiac death in a population of older adults. J Am Coll Cardiol. 2006;47:362-7.
  2. Kimura Y, Okumura T, Morimoto R, Kazama S, Shibata N, Oishi H, Araki T, Mizutani T, Kuwayama T, Hiraiwa H, Kondo T, Murohara T. A clinical score for predicting left ventricular reverse remodelling in patients with dilated cardiomyopathy. ESC Heart Fail. 2021;8:1359-1368.
